# *Chlamydomonas reinhardtii* Alternates Peroxisomal Contents in Response to Trophic Conditions

**DOI:** 10.3390/cells11172724

**Published:** 2022-09-01

**Authors:** Naohiro Kato, Clayton McCuiston, Kimberly A. Szuska, Kyle J. Lauersen, Gabela Nelson, Alexis Strain

**Affiliations:** 1Department of Biological Sciences, Louisiana State University, Baton Rouge, LA 70803, USA; 2Bioengineering Program, Biological and Environmental Sciences and Engineering Division, King Abdullah University of Science and Technology (KAUST), Thuwal 23955-6900, Saudi Arabia

**Keywords:** *Chlamydomonas*, microalgae, fluorescence microscopy, peroxisomes, fatty acids, β-oxidation, glyoxylate cycle enzymes, PEX, PTS1

## Abstract

*Chlamydomonas reinhardtii* is a model green microalga capable of heterotrophic growth on acetic acid but not fatty acids, despite containing a full complement of genes for β-oxidation. Recent reports indicate that the alga preferentially sequesters, rather than breaks down, lipid acyl chains as a means to rebuild its membranes rapidly. Here, we assemble a list of potential *Chlamydomonas* peroxins (PEXs) required for peroxisomal biogenesis to suggest that *C. reinhardtii* has a complete set of peroxisome biogenesis factors. To determine involvements of the peroxisomes in the metabolism of exogenously added fatty acids, we examined transgenic *C. reinhardtii* expressing fluorescent proteins fused to N- or C-terminal peptide of peroxisomal proteins, concomitantly with fluorescently labeled palmitic acid under different trophic conditions. We used confocal microscopy to track the populations of the peroxisomes in illuminated and dark conditions, with and without acetic acid as a carbon source. In the cells, four major populations of compartments were identified, containing: (1) a glyoxylate cycle enzyme marker and a protein containing peroxisomal targeting signal 1 (PTS1) tripeptide but lacking the fatty acid marker, (2) the fatty acid marker alone, (3) the glyoxylate cycle enzyme marker alone, and (4) the PTS1 marker alone. Less than 5% of the compartments contained both fatty acid and peroxisomal markers. Statistical analysis on optically sectioned images found that *C. reinhardtii* simultaneously carries diverse populations of the peroxisomes in the cell and modulates peroxisomal contents based on light conditions. On the other hand, the ratio of the compartment containing both fatty acid and peroxisomal markers did not change significantly regardless of the culture conditions. The result indicates that β-oxidation may be only a minor occurrence in the peroxisomal population in *C. reinhardtii*, which supports the idea that lipid biosynthesis and not β-oxidation is the primary metabolic preference of fatty acids in the alga.

## 1. Introduction

Photosynthetic microalgae are unicellular microbes that can consume carbon dioxide as their sole carbon source for growth and are powered by light energy. Microalgae can produce considerable amounts of triacylglycerols (TAGs) under stressful environmental conditions such as nitrogen depletion [1] he regulation of TAG and fatty acid (FA) syntheses in microalgae has been studied relatively well because of their potential as a source of biofuels [2,3]. The model green microalga *Chlamydomonas reinhardtii* has long served as a model organism to study the regulation of lipid metabolic pathways in photosynthetic organisms, although it is not an oleaginous organism [4] Its role as a model alga is due to its relatively simple cellular structure, available molecular tools, and ease of handling. *C. reinhardtii* can grow autotrophically with carbon dioxide and light, heterotrophically with acetic acid (C2), and mixotrophically with combinations of both. In the alga, acetic acid is first metabolized by conversion to acetyl-CoA. Through the concerted action of enzymes of the glyoxylate cycle, acetyl-CoA is converted to succinate (C4) and subsequently channeled into the gluconeogenesis [5]. The enzymes required for this conversion, except isocitrate lyase, have been localized to the peroxisomes [6].

FA catabolism is an important and well-studied metabolic process in animals and land plants for membrane turnover and energy generation [7]. FA catabolism begins with enzymatic lipases catalyzing the hydrolysis of TAGs (lipolysis) to yield free FAs and glycerol. In land plants, free FAs are subjected to β-oxidation to break down acyl chains into acetyl-CoA units, which then enter the glyoxylate cycle as above or by the citric acid cycle for respiratory energy generation [8]. Catabolism of FAs, therefore, allows the breakdown of lipids into 2-carbon molecules to be utilized as energy and carbon building blocks in cells. This process is essential for land plants because it allows seeds to germinate and grow without light. Whereby acyl-chains serve as a long-term energy source in addition to starches [9].

β-oxidation and the glyoxylate cycle occur within the peroxisomes in land plants [10]. These organelles handle many aspects of overflow metabolism and crosstalk and possess heterogenous biochemistries across different species and organisms [11,12]. Peroxisome biogenesis, which requires a set of PEX (peroxin) structural proteins and protein import into the peroxisomal matrix, is mediated by either C-terminal peroxisomal targeting (PTS) 1 or N-terminal PTS2 sequences [13]. Although PTS1/2 sequences mediate the majority of peroxisomal import, there is evidence of proteins being carried in complexes and non-PTS mediated import [14]. Dual targeting of proteins in other organelles and peroxisomes has also been observed from PTS2 (N-terminal) signals, and it is challenging to predict localization bioinformatically [15]. The known canonical sequences found in plants and animals have recently been observed to be flexible in *C. reinhardtii*, where the catalase isoform 1 (CAT1) [16] and aconitate hydratase [6] are found to have non-canonical PTS1.

The diversity in sequence targeting possibilities and difficulty in the bioinformatic prediction of peroxisomal targeting in algal cells has left gaps in our knowledge of how these organelles support metabolic functions in the alga. Of the various metabolic processes in which these organelles could be involved, much remains unknown in *C. reinhardtii*. A study suggests that exogenously added FAs suppress the growth of the algae in illuminated conditions [17,18]. We previously observed that exogenously added FAs are incorporated into microbodies within 10 min [16] and metabolized in 24 h [19]. The microbodies resembled those observed in peroxisomal localized glyoxylate cycle enzymes [6]. Although the transport mechanism of the exogenously added FA into *C. reinhardtii* is not clearly understood, the incorporation is thought to occur by a mechanism similar to yeast that utilizes a FA transport protein with a long-chain acyl-CoA synthetase [16]. We also recently found that these two microbodies were not the same and that exogenous FAs accumulated into fatty acid-induced microbodies (FAIMs), while the PTS2 containing citrate synthase 2 (CIS2) glyoxylate cycle enzyme exhibited a different localization [16]. Yet, the structures of these organelles appear identical in transmission electron microscopy [17,19]. Moreover, Kong et al. identified the β-oxidation activity in the *Chlamydomonas* peroxisomes [18]. Hence, one may assume Chlamydomonas can grow on exogenously added FAs through the incorporation into the peroxisomes, where β-oxidation and glyoxylate cycle occur to convert the FAs to acetyl-CoA.

Here, we characterized the contents of the peroxisomes using transgenic *C. reinhardtii* expressing fluorescent proteins fused to N- or C-terminal peptide of peroxisomal enzymes, concomitantly with fluorescently labeled palmitic acid, under different growth conditions. We used confocal microscopy to track the populations of the peroxisomes in illuminated and dark conditions with and without acetic acid as a carbon source. Furthermore, we assembled a list of potential PEX proteins to suggest that *C. reinhardtii* has a complete set of peroxisome biogenesis factors. Yet, we found that the alga is unable to utilize exogenously added FA for heterotrophic growth. Our findings also indicate that light conditions alter the peroxisomal contents in *C. reinhardtii*.

## 2. Materials and Methods

### 2.1. Algal Strains and Culture Conditions

*C. reinhardtii* strains CC4425 (cell wall deficient mutant, *nit2^−^, cw15, mt^+^*), and CC124 (wild type, *mt^−^*) were obtained from the Chlamydomonas Resource Center. *Chlamydomonas* strain UVM4 was provided by Prof. Ralph Bock [20]. *C. vulgaris* (UTEX#395) was obtained from the Culture Collection of Algae at the University of Texas at Austin. The strains were maintained in 250 mL flasks containing 100 mL liquid tris-acetate-phosphate medium (TAP) or on agar plates [21]. The cultures were kept at 23 ± 2 °C under fluorescence light (60 μmol/m^2^/s) and constantly shaken on an orbital shaker at 180 rpm. Two milliliters of three-day-old-cultured cells were transferred from the flasks to test tubes to examine changes in cell densities under different culture conditions. Culturing on agar plates was conducted by streaking the algae in the flasks on plates that contain 1.5% (*w*/*v*) agar.

### 2.2. Fatty Acid Preparations

Fatty acid solutions for the *C. reinhardtii* culture medium were prepared as 75 mM butyric acid (C4 FA, BTA), 75 mM valeric acid (C5 FA, Val A), or 0.5% (*w*/*v*) Tween 80, which were directly added to MM (TAP without acetate). When *C. reinhardtii* and *Chlorella vulgaris* were cultured on agar plates, 0.5% (*v*/*w*) Tween 80 was first added to the autoclaved agar medium, then 5 mM palmitic acid or 5 mM oleic acid was added directly into the medium. When *Chlamydomonas* CC5082 was cultured with different carbon sources in test tubes, 5 mM palmitic acid, 5 mM oleic acid, or 7 mM glucose were added in the test tubes with 0.5% (*v*/*w*) Tween 80. For fluorescent labeling experiments, 10 mM BODIPY^®^ FL-C16 (Life Technologies, hereafter FL16, Agawam, MA, US) was prepared by diluting 1 mg BODIPY^®^ FL-C16 with 210 μL of DMSO. FL-16 was added to cultures at a ratio of 200 μL to 100 mL of TAP (final concentration 20 μM) with 100 μM of oleic acid.

### 2.3. Fluorescent Protein Constructs and Algal Transformation

Transformation of UVM4 was conducted by the glass-bead method as previously described [22], and colonies were selected with paromomycin (10 mg L^−1^) and zeocin (15 mg L^−1^), depending on the desired combination of constructs. All constructs were generated with pOpt_2 vectors for each respective fluorescent marker: mCerulean3 (CFP) or mRuby2 (RFP) [23]. CFP constructs were generated with a paromomycin resistance cassette, while RFP constructs were generated in a bleomycin/zeocin resistance vector to enable double transformations and co-localization experiments. Screening for expression of each targeted marker was done by stereo microscopy on the agar-plate level as previously described [24], and several transformants for each construct were generated before confocal microscopy. Targeting peptides for citrate synthase (CIS2) and the serine-lysin-isoleucine tripeptide (SLI) were taken from our previously described plasmid constructs [6,24].

### 2.4. Fluorescence Microscopy 

For all cells analyzed by fluorescence microscopy, 1 µL aliquots of cultures were dropped on a microscope slide and covered by a cover glass with a spacer. The cells were then viewed with a White Light Laser Confocal Microscope Leica TCS SP8X equipped with a ×63 (N.A. 1.20) water immersion lens. Three optical conditions were set so that the spectra of fluorescent probes within cells were individually resolved. The optical conditions are shown in Table 1.

### 2.5. Compartment Analysis

The acquired images were analyzed with Fiji, the image processing software [25] with its plugin ComDet v.0.5.5 (https://github.com/ekatrukha/ComDet). Optical sections that contain at least one small compartment (>2 mm diameter) for each FL-C16, CIS2-RFP, and CFP-SLI, respectively, were analyzed. Maximum distance between colocalized spots was set as 5 pixels.

### 2.6. Identification of PEX Genes

Keyword searches with PEX as a query using the Phytozome v5.5 database (https://phytozome-next.jgi.doe.gov) were first conducted to determine a complete set of peroxisome biogenesis genes was present in the genome of *C. reinhardtii*. BLAST searches were then done with PEX sequences of *Arabidopsis thaliana* [13], *Drosophila melanogaster*, *Hansenula polymorpha*, *Homo sapiens*, *Mus musculus*, *Saccharomyces cerevisiae*, *Yarrowia lipolytica,* or *Neurosporacrassa* [26] against *Chlamydomonas* protein database with default search parameters. Blast hits with the except value smaller than e^−5^ were then identified as a homologous gene.

## 3. Results

3.1.C. Reinhardtii Cannot Grow Heterotrophically on Fatty Acids 

Although *C. reinhardtii* can grow on acetic acid (C2) heterotrophically, fatty acid (FA) driven growth remains a debatable topic [17,18]. Here, we tested whether *C. reinhardtii* could grow heterotrophically on exogenously added FAs of various carbon chain lengths (Figure 1). 

After pre-culturing cells in a standard tris-acetate-phosphate (TAP) medium that contained 7 mM acetic acid, they were transferred to a minimal medium (MM) [21] that contained only nutrients necessary for autotrophic growth. Alternatively, the pre-cultured cells were transferred to MM containing different short-chain FAs. Their growth profiles were then investigated. Because long-chain FAs are hydrophobic, they require a dispersing agent to be soluble in the culture medium; for this, we used Tween 80, which has been previously shown to be effective for this purpose in other microbial studies [27,28]. Butyric acid (C4 FA, BTA), valeric acid (C5 FA, Val A), or Tween 80 alone were added to MM instead of acetic acid (C2) (Figure 1A). Here, we tested the standard laboratory strain *C. reinhardtii* CC124, CC4245, and the genome sequence-verified wild-type strain CC5082 to ensure that the growth observed was not a strain-specific phenomenon. The strains were cultivated in illuminated (light) or dark conditions in these media. Both strains proliferated during the 3-day cultivation when grown with light regardless of medium components. FA or Tween 80 addition did not increase growth compared to MM conditions. However, only cells cultivated in a standard acetic acid-containing TAP medium proliferated when the strains were cultivated in the dark. Cell densities did not increase during the cultivations in the dark with MM, BTA, ValA, or Tween (Figure 1A). 

We also examined whether *C. reinhardtii* could use exogenously added palmitic acid (C16:0 FA) for its proliferation, as palmitic acid is a major component of the algal polar and neutral lipid fractions [29] (Figure 1B). We also tested *Chlorella vulgaris* (UTEX#395), a Chlorophyta member that can grow with other carbon sources such as glucose. The strains were streaked on agar containing acetic acid or palmitic acid, again with Tween 80 as a dispersing agent, and placed in the dark for 20 days. For both algal species tested, growth in the dark was detected with acetic acid. However, growth was not detected with palmitic acid (Figure 1B). Furthermore, we analyzed whether the CC5082 strain can grow with palmitic acid in a liquid medium (Figure 1C). CC5082 was not capable of growth on palmitic acid as a carbon source in a liquid medium. However, it could tolerate it when acetic acid was present or when the culture was illuminated (Figure 1C). With oleic acid (C18:1 FA), CC5082 died (bleached out) in a liquid medium both with and without illumination under our conditions (Appendix A).

### 3.2. C. Reinhardtii Has a Complete Set of Peroxisome Biogenesis Factor Genes

In *C. reinhardtii*, orthologs of the β-oxidation enzymes have been previously revealed [18,30,31,32]. As *C. reinhardtii* was not able to grow on externally added FAs heterotrophically, we were curious as to whether the formation of the algal peroxisomes was in some way incomplete compared to yeasts and land plants. Previous analysis conducted by Smith and Aitchison curated the peroxisomal biogenesis (PEX) genes of common eukaryotic model organisms and several fungi [13], while Nito et al. thoroughly described those of *Arabidopsis thaliana* [26]. To our knowledge, no curation of these genes in *C. reinhardtii* has been performed. Therefore, we used available genomic resources to conduct investigations of the algal genome with basic local alignment search tools (BLAST). We searched the *C. reinhardtii* genome for these PEX genes using a combination of keyword searches as well as *A. thaliana (At)* and *Saccharomyces cerevisiae* (*Sc*) PEX amino acid sequences (Phytozome v5.5) (Table 2). *C. reinhardtii* was found to contain all PEX genes which are present in the *A. thaliana* genome. *Cr*PEX3 was not annotated and could not be directly found by BLAST of *At* or *Sc* PEX3. This is a crucial enzyme involved in incorporating proteins into the peroxisomal membrane, and its absence results in peroxisome biogenesis complecation in plants and yeasts [33]. However, BLAST of *At* or *Sc* PEX3 genes against Chlorophyceae in the National Center for Biotechnology Information (NCBI) database revealed a hit in the closely related alga *Volvox carteri.* When this protein sequence was used again to search in the *C. reinhardtii* genome, Cre14.g618450 was found, which is annotated as a non-specific serine/threonine-protein kinase. However, the analysis of the peptide domain encoded in Cre14.g618450 with InterPro [34] identified an integral component of the peroxisomal membrane (GO:0005779) but not the kinase domain. This supports the idea that *C. reinhardtii* encodes PEX3 in the genome. We also found that all of the *C. reinhardtii* PEX genes were expressed in various culture conditions (Appendix A and Appendix A).

### 3.3. C. Reinhardtii Contains Peroxisomes with Varying Contents

We previously found that exogenously added FAs are incorporated in microbody-like compartments that structurally resembled (containing a lipid-bilayer) peroxisomes in *C. reinhardtii* and named these structures FAIMs [19]. We found that FAIMs do not accumulate the glyoxylate cycle enzyme citrate synthase 2 (CIS2) [16], which was previously shown to localize in the same peroxisomes as *Cr*ACX2, an essential enzyme of the β-oxidation cycle [18]. To test whether the FAIMs accumulate any other peroxisomal protein, we generated simultaneously a *C. reinhardtii* transgenic strain expressing mRuby2 (red fluorescent protein, hereafter RFP) fused to the 25 amino acid sequence of the N-terminal end of the CIS2 protein (CIS2-RFP) and cyan fluorescent protein (hereafter CFP) fused to synthetic C-terminal PTS1, a serine-lysine-isoleucine tripeptide [S][L][I] (CFP-SLI). The previous study identified PTS1 tripeptide in land plants that share the mechanism of peroxisomal import with *C. reinhardtii* (Table 2) [14]. The study revealed that the functional plant PTS1 tripeptides are x[KR][LMI], [SA]y[LMI], or [SA][KR]z (x, y, z is a low abundant amino acid). The [S][L][I] tripeptide is categorized as the functional peptide in land plants, and a previous study showed that the mVenus-SLI protein is localized in the peroxisomes in *C. reinhardtii* [24]. Confocal microscopy of these two independent markers with fluorescently labeled palmitic acid (hereafter FL-C16) feeding found two different patterns of accumulations in the peroxisomes (Figure 2).

The first pattern was the accumulation of CIS2-RFP and CFP-SLI but not FL-C16. The second pattern was the accumulation of FL-C16 and CFP-SLI but not CIS2-RFP. This result suggested that exogenously added fatty acid would accumulate in the peroxisomes where PTS1-containing peroxisomal proteins are accumulated, but not glyoxylate cycle enzymes. 

To understand the dependency of differential accumulation patterns of the fluorescent markers in more detail, we conducted a pulse experiment using the same transgenic strain. We previously found that changes in fluorescence intensity of FL-C16 reached equilibrium at 24 h, and then metabolized or maintain the equilibrium status up to 72 h in the pulse experiment [19]. In this experiment, FL-C16 was first fed for 8 h while the cells were cultured in illuminated conditions, and then the cells were washed and cultured in four different conditions for 24 h. The conditions were (1) TAP Light in which both acetic acid and light are available (mixotrophic condition), (2) MM Light in which light is available (autotrophic condition), (3) TAP Dark in which acetic acid is available (heterotrophic condition), and (4) MM Dark in which no exogenous energy is available. We then quantitatively analyzed colocalizations on optically section images (Figure 3).

The analysis suggested that the total number of compartments (bright intensity spots with either FL-C16, CIS2-RFP, or CFP-SLI) is reduced by about 40% when the cells are cultured without acetic acid (MM) regardless of the light conditions (0.59-fold change in an average [*p* = 6.98 × 10^−7^] when the cells are cultured in MM Light, compared to when the cells are culture in TAP Light. A 0.57-fold change in an average [*p* = 0.002] when the cells are cultured in MM Dark, compared to when the cells are cultured in TAP Dark). The number of compartments that contain FL-C16, CIS2-RFP, or CFP-SLI is equally reduced (Appendix A). On the other hand, we identified all possible co-localization patterns in the compartments, including those containing only one marker (Figure 3C). The compartments in which CSI2-RFP and CFP-SLI are colocalized but not FL-C16 (≥21% of the population) and the compartments in which FL-C16 alone is localized (≥28% of the population) are the major types of the compartments in all cultural conditions. The compartments that contain all three markers (FL-C16, CSI2, and SLI) also are detected in all culture conditions, although they are rare (<3% of the compartment population). FL-C16 is co-localized with CFP-SLI in about 5% of the compartment population when the cells are cultured with acetic acid (TAP) regardless of light conditions. The compartments containing only glyoxylate cycle marker, CIS2-RFP, increase about 2-fold when the cells are grown in the dark, compared to when the cells are grown in light, irrespective of the acetic acid status in the culture medium (from 6% to 13% in TAP, and from 13% to 28% in MM). These results suggest that *C. reinhardtii* carries different types of peroxisomes in the same cell, and the cultural conditions alter the ratio of the peroxisomal population. The exogenously added fatty acids are rarely compartmentalized with the peroxisomal markers tested in this study.

## 4. Discussion

4.1.C. Reinhardtii Does Not Grow on Fatty Acids 

Currently, available models in systems biology assume *C. reinhardtii* catabolizes TAG and FA for biosynthesis and energy [35]. However, this depends solely on assumptions that transcriptional changes in lipase and β-oxidase-related genes correlate to their predicted functions. However, a recent flux analysis using isotopic acetic acid found that the primary purpose of TAG and FA catabolism in alga is to form membrane lipids rather than energy production [36]. We found that *C. reinhardtii* cannot grow heterotrophically on exogenously added C4, C5, and C16 FAs (Figure 1). A previous study described heterotrophic growth of *C. reinhardtii* with oleic acid (C18:1) [18]. However, the growth was tracked for only 48 h, and the difference in cell densities between oleic acid and heterotrophic ethanol growth (negative control) was only 15% in one strain. Similar differences between FAs and Tween 80 (FA dispersion agent) were observed in our experiments at 48 h (Figure 1). In our culture conditions, oleic acid is somewhat toxic (Appendix A), suggesting that exogenously added oleic acid cannot sustain the algal growth or may be harmful for long time cultivation. 

We previously determined that most glyoxylate cycle enzymes, including malate dehydrogenase 2 (MDH2) and acetyl-CoA synthase, accumulate in the peroxisomes in *C. reinhardtii* [6]. This localization was observed with both peroxisome targeting peptide classes (PTS1 and PTS2), which use different import machinery (PEX5 and PEX7, respectively) [37]. Another study found that alternative oxidase-2 (*Cr*ACX2), a vital enzyme of the β-oxidation pathway, co-localizes with MDH2 in the peroxisomes in *C. reinhardtii* [18]. In addition, we recently observed that catalase isoform 1 (CAT1) localizes in the peroxisomes mediated by a non-canonical PTS1 targeting peptide [16]. These previously reported findings imply that the peroxisomes of *C. reinhardtii* should be capable of many of the roles ascribed to them in land plants and yeasts, such as β-oxidation and glyoxylate cycle. Inability to grow on exogenously added FAs despite having a full complement of gene homologs dedicated to these processes in other organisms suggests that other differences, such as compartmentalization of the gene products, exist in the algal cell, which prevents heterotrophic growth on FAs.

We found that the total number of compartments that contain CIS2-RFP, CFP-SLI, or FL-C16 is reduced when the cells are cultured without acetic acid (MM) regardless of the light conditions (Figure 3C). This finding agrees with the previous report that the number of peroxisomes accumulating GFP-SRL (GFP fused to PTS1 tripeptide) is higher when *C. reinhardtii* is cultured with acetic acid compared to when the cells are cultured with ethanol or sucrose [17]. It also agrees with the gene expression data (Appendix A and Appendix A). The data suggest that all *C. reinhardtii* PEX genes, except PEX11 and PEX27, are constantly expressed at low levels in various culture conditions. The PEX11 and PEX27 genes are expressed significantly higher than the rest of the PEX genes in the various culture conditions (*p* < 0.001 in each pair Student’s *t*-test), and the expressions are further upregulated when the cells are cultured with acetic acid. PEX11 and PEX27 in yeast are involved in the formation and fission of the peroxisomes [38]. Hence, our study and the previous report support the idea that the biogenesis of peroxisomes in *C. reinhardtii* would be upregulated by acetic acid. Interestingly for us, the number of the compartments containing FL-C16 (FAIMs) is significantly increased with acetic acid, mean 1.8 to 4.2 [*p* < 0.0001] in the illuminated condition; mean 2.6 to 4.8 [*p* = 0.0007] in the dark condition) (Appendix A). This suggests that the FAIMs may be the peroxisomes or share biogenesis machinery for their formation.

### 4.2. Physical Separation of Exogenously Added Fatty Acids and Enzymes for β-oxidation and Glyoxylate Cycle

Enzymes involved in β-oxidation should be localized in organelles where they can act on FA acyl groups to generate acetyl-CoA units from these hydrocarbons. Here, we observed that about 5% of the compartments contained FL-C16 and CIS2-RFP or CFP-SLI regardless of the culture conditions (Figure 3D). The compartments containing FL-C16 were originally identified as FAIMs in the study of lipolysis in *C. reinhardtii* [19]. The FAIMs were physically separated from the compartments containing CIS2-CFP when the cells were cultured in TAP with an illumination (mixotrophic condition) [16]. This study analyzed the co-localization in different cultural conditions. The results indicate that a limited population of the FAIMs import the peroxisomal markers independently of the trophic conditions. As it was discussed above, this suggests that the FAIMs may be the peroxisomes or share biogenesis machinery for their formation. We previously showed that FL-C16 accumulates in the lipid droplets in *C. reinhardtii* when the cells are cultured in nitrogen-deficient conditions [16]. Together with the results in this study, we hypothesize that the FAIMs may represent a mixed population of lipid droplets and peroxisomes. Alternatively, we hypothesize that the FAIMs are the peroxisomes into which different sets of peroxisomal proteins not examined in this study are selectively imported. Biosynthesis and the structure of the peroxisomes and lipid droplets from the endoplasmic reticulum in *C. reinhardtii* are not clear [39]. It is required to systematically analyze localizations in expected peroxisomal proteins and marker proteins for the peroxisomal membrane proteins, lipid droplet membrane proteins, and endoplasmic reticulum proteins to conclude the change of peroxisomal and lipid droplet contents by different culture conditions. Nonetheless, our data suggest that β-oxidation may be only a minor occurrence (about 5%) in the peroxisomal population in *C. reinhardtii*, which agrees with the previously conducted metabolic analysis indicating that storage of acyl chains but not β-oxidation is the primary metabolic preference of the alga [36]. 

### 4.3. Changes in Peroxisomal Contents by Light Conditions

We found that the rate of the compartment containing only CIS2-RFP in the cell increases about 2-fold regardless of the acetate conditions when the cells are cultured in the dark (Figure 3D). When acetic acid is in the cultured medium (TAP), the total numbers of the compartments, as well as the numbers of individual compartments (i.e., containing FL-C16, CIS2-RFP, or CFP-SLI), are not significantly different between the illuminated and dark conditions (Figure 3D and Appendix A). The expressions of the PEX11 and PEX27 genes that are upregulated by acetic acid are also not alternated by the light conditions (Appendix A and Appendix A), supporting the idea that the biogenesis of peroxisomes is not regulated by light. Rather, this suggests that distributions of CIS2-RFP are altered among the compartments by the light conditions. On the other hand, when acetic acid is not in the medium (MM), the total number of the compartments is not significantly different between the illuminated and dark conditions, but a significant decrease in the number of the compartments containing CFP-SLI only is observed in the dark condition (mean 3.3 to 1.8 [*p* = 0.0069]) (Appendix A). A decrease in the rate in the compartment containing CFP-SLI only (17% to 8%) as well as a decrease in the rate in the compartment containing CIS2-RFP2 and CFP-SLI (36% to 21%) are also observed (Figure 3D). These suggest that while distributions of CFP-SLI are altered among the compartments, CFP-SLI may also be degraded in the dark. We identified 443 genes that encode the PTS1 tripeptides (x[KR][LMI], [SA]y[LMI], or [SA][KR]z) that would be recognized by *Chlamydomonas* PEX5 for the peroxisomal import in the *Chlamydomonas* genome (Appendix A). Of these, six genes encode SLI. Further analysis besides microscopy is required to examine a hypothesis in which proteins that encode specific PTS1 selectively alter the organellar localization and the protein stability in the dark. Moreover, protein localizations determined by this work were conducted by expressing recombinant proteins in the UVM4 stain developed for high expression of recombinant proteins [40]. Hence, the results obtained in this study may need to be confirmed using the native proteins in wild-type *C. reinhardtii.* Nevertheless, these observations suggest that the peroxisomal contents are altered by light conditions in *C. reinhardtii*. When *C. reinhardtii* is in the dark, photosynthesis and photorespiration are not activated, yet increases in expressions of mRNAs and proteins that upregulate photosynthesis and photorespiration are observed [41]. The alteration of the peroxisomal contents may contribute to balancing carbon flux by altering the FA metabolism in illuminated and dark conditions. 

### 4.4. Peroxisomal Function and Evolution in Green Algae

When *C. reinhardtii* is grown for about 200 generations in continuous light without acetic acid, it loses its ability to grow heterotrophically in the dark on acetic acid [42]. This implies a rapid adaptation of *C. reinhardtii* to environmental conditions. In prolonged darkness immobile land plants cannot rely on photosynthesis for energy, and the breakdown of FA for survival is required, for example, in the germination and growth of seeds [43]. However, the flagellate (motile) algae exhibit phototaxis and can potentially move to light sources for energy generation [44]. Phototaxis may have meant that these organisms did not need to rely on β-oxidation for energy generation in the dark, which may have been lost in their evolution. We found the non-flagellate alga *Chlorella vulgaris* is also incapable of heterotopic growth on palmitic acid (Figure 1), suggesting that this may be a feature of the Chlorophyta division, which originally possesses flagella. *Chlorella* belongs to the division but is believed to lose flagella during evolution [45,46]. Investigating the relationship between cellular motility and β-oxidation capability in green algae is worth further investigation. 

## 5. Conclusions

The acetate-flagellate microalga *C. reinhardtii* has served as a model organism for generations, yet only recently have peroxisomes been described in this alga, and their metabolic implications have become increasingly discussed [16,18,47]. Although *C. reinhardtii* contains all genetic elements necessary for β-oxidation and formation of peroxisomes, it seems incapable of growth on endogenous or exogenous FAs as energy sources. Here, we demonstrated that part of this contradiction may be a spatial separation of the FA processes and the peroxisomal enzymes. Numerous questions have arisen from these investigations: how does the alga regulate protein import into the peroxisomes, how is the peroxisome spatial separation regulated, and has the capacity for phototaxis led *C. reinhardtii* (and other algae) to evolve efficient lipid recycling mechanisms that favor membrane biogenesis while circumventing energy generation from an acyl-chain breakdown? With the results presented in this work, we illustrate the dynamic nature of peroxisome contents in the algal cell and suggest their role in FA metabolism. 

## Figures and Tables

**Figure 1 cells-11-02724-f001:**
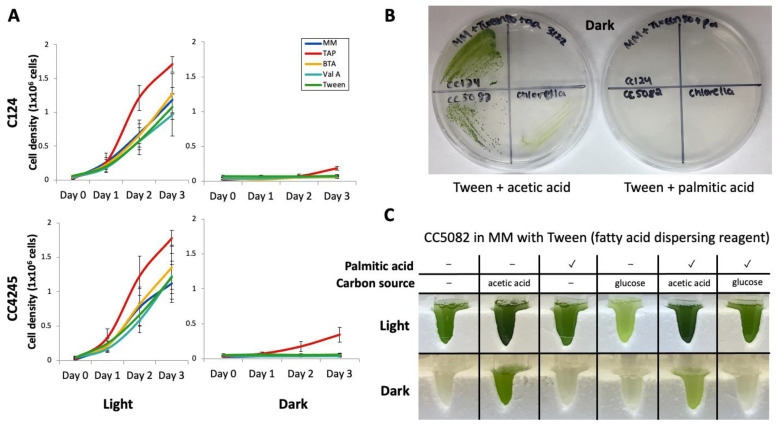
*C. reinhardtii* is not able to grow on exogenous fatty acids. (**A**) *C. reinhardtii* culture cell density after 3 d following transfer from TAP medium to minimal medium (MM), TAP, or MM with the addition of butyric acid (C4, BTA) in Tween 80 (FA dispersing agent, Tween), valeric acid (C5, Val A) in Tween 80, or Tween 80 alone. About 5 × 10^6^ cells of strain CC124 or CC4425 were transferred to each new medium for each experiment. The graph shows averages ± standard error mean of three independent experiments. (**B**) Growth of *C. reinhardtii* and *C. vulgaris* cultured on agar plates without light. *C. reinhardtii* CC124, CC5082, and *C. vulgaris* (UTEX#395) were cultured for 4 d in TAP, then streaked on agar plates containing either acetic acid (C2 FA, left) or palmitic acid (C16 FA, right) and placed in the dark for 20 d. Tween is a dispersant for palmitic acid but did not affect growth with the acetic acid present. Growth was not observed when C16 FA was used as a sole carbon source in the dark (right Petri plate). (**C**) Growth of *Chlamydomonas* CC5082 cultured with different carbon sources. CC5082 was cultured for 14 days in liquid MM that contained Tween and other carbon sources, such as acetic acid, palmitic acid, and glucose. The cells were cultured with or without light. Notice that the cells proliferate in the dark only when acetic acid is in the medium.

**Figure 2 cells-11-02724-f002:**
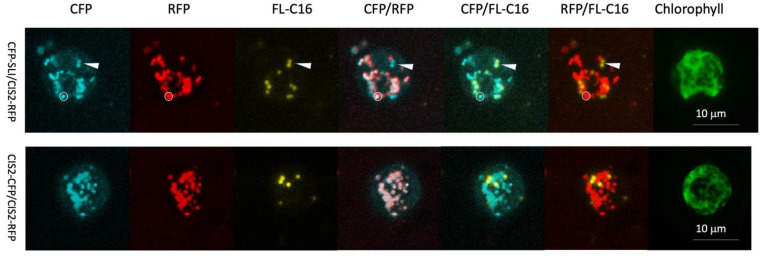
Co-localization of FL-C16 and CFP-SLI (PTS1 containing peroxisomal targeting protein) was identified in *C. reinhardtii*. Accumulations of FL-C16 and peroxisomal marker proteins after the cells were cultured in TAP for 24 h in illuminated conditions. Thirty optical sections of each 20 µm z-stack are projected into a plane image with maximum intensity. CFP-SLI: CFP fused to the PTS1 signal peptide serin-leucine-isoleucine. CIS2-RFP: Twenty-five N-terminal amino acid sequences of glyoxylate cycle enzyme citrate synthase 2 (CIS2) fused to RFP. CIS2-CFP: CIS2 fused to CFP. A compartment that contains CIS2-RFP and CFP-SLI but not FL-C16 is shown with a white circle. A pair of compartments that contain FL-C16 and CFP-SLI but not CIS2-RFP are shown with an arrowhead (upper panel). Notice that CIS2-CFP and CIS2-RFP are all colocalized (lower panel), suggesting that non-colocalization between CFP-SLI and CIS2-RFP (marked with the arrowhead) is not an artifact by protein expression or imaging acquisition.

**Figure 3 cells-11-02724-f003:**
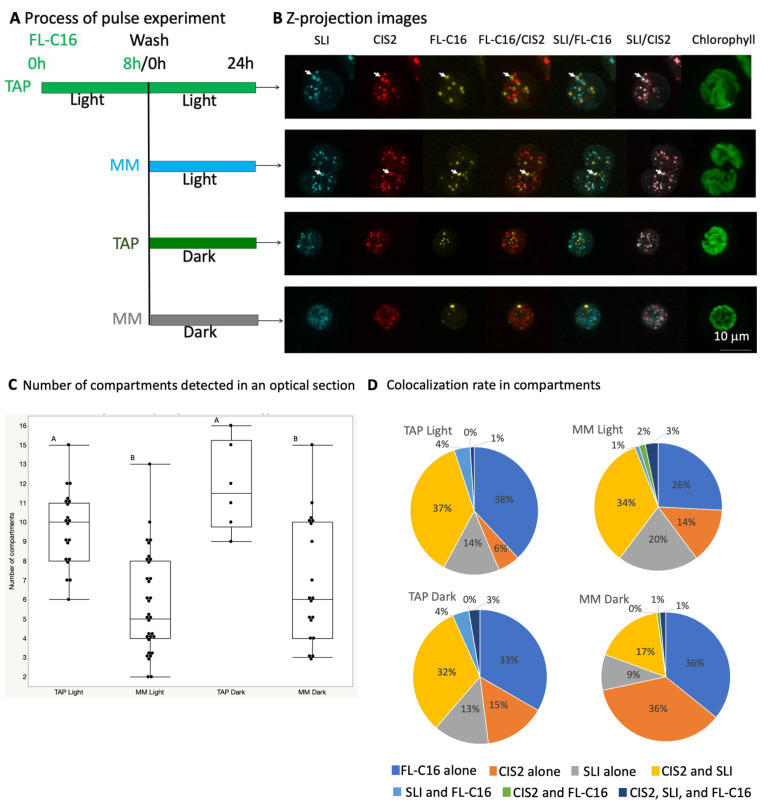
Statistical analysis revealed alteration of peroxisomal content by light conditions. (**A**) Schematic of fluorescently labeled palmitic acid (FL-C16) pulse experiment. *C. reinhardtii* expressing both CFP-SLI and CIS2-RFP was cultured in TAP with FL-C16 for 8 h, then transferred to a fresh medium under four different conditions for 24 h before imaging: TAP with continuous light, minimal medium (without acetate, MM) with continuous light, TAP in the dark, and MM in the dark. (**B**) Localization of FL-C16 and peroxisomal marker proteins after 24 h in each condition. Thirty optical sections of each 20 µm z-stack are projected into one image with their maximum intensity. Co-localizations of all three markers are rarely observed, indicated by white arrows. (**C**,**D**) Thirty optical sections within the 20 µm z-axis were obtained by confocal microscopy. Colocalization in the optical sections was analyzed with ComDet (ImageJ plugin for analyzing colocalization of bright intensity spots). Mean of the spot number, standard error of the mean, numbers of cells, and optical sections for the analysis are, TAP with continuous light (TAP Light: mean 9.7, S.E. 0.47, 8 cells, 20 sections), MM with continuous light (MM Light: mean 5.75, S.E. 0.46, 7 cells, 32 sections), TAP in the dark (TAP Dark: mean 12.1, S.E. 2.78, 3 cells, 6 sections), and MM in the dark (MM Dark: mean 6.94, S.E. 0.76, 5 cells, 19 sections). (**C**) Compartments detected in an optical section are reduced when the cells are cultured without acetic acid. A total number of compartments in an optical section detected by ComDet is presented as a box plot with individual data points. Notice that the number of compartments is reduced when the cells are cultured without acetic acid (MM) regardless of light conditions. A letter on the top of each bar indicates mean comparisons for each pair using Student’s *t*-test. Levels not connected by the same letters are significantly different (*p* < 0.05). (**D**) Compartments change contents depending on culture conditions. Colocalization rates in the compartments are calculated, based on the culture conditions (TAP Light: *n* = 194, TAP Dark: *n* = 105, MM Light: *n* = 73, MM Dark, *n* = 132).

**Table 1 cells-11-02724-t001:** The optical conditions.

	Excitation (nm)	Emission (nm)
CFP	458	468–500
FL-C16	500	510–530
RFP	555	562–591
Chlorophylls	458	697–800

The cells were imaged at 0.65 μm z-step intervals for 30 steps.

**Table 2 cells-11-02724-t002:** The genome of *C. reinhardtii* contains a complete set of peroxisome biogenesis genes. * Hit derived from *At* PEX3-1 BLAST against Chlorophyceae in the NCBI database identified Vocar.0001s1462.1, the gene in *Volvox carteri*. A BLAST search with this gene against *Cr* identifies Cre14.g618450. *Yl* PEX9 is a homolog of PEX26 from other organisms. ** Farré et al. 2017, PEX36 homolog of PEX16 in *Komagataella phaffii* (formerly *Pichia pastoris*) but absent from *S. cerevisiae.* *** Function as PEX17 questioned by [26]. No hits in *Cr* for *At*4g18197, *At* PEX17, was renamed purine permease 7 (PUP7) and is likely not an orthologue of *Sc* PEX17. *Hanseula.*

PEX Genes	Saccharomyces Cerevisiae	Yarrowia Lipolytica	Hansenula (Ogataea) Polymorpha	Neurospora Crassa	Drosophila Melanogaster	Mus Musculus	Homo Sapiens	Arabidopsis Thaliana	Chlamydomonas Reinhardtii	Phytozome Accession	Description
PEX 1	✓	✓	✓	✓	✓	✓	✓	✓	✓	Cre16.g679200	peroxisome biogenesis factor 1
PEX 2	✓	✓	✓	✓	✓	✓	✓	✓	✓	Cre17.g698800	predicted e3 ubiquitin ligase
PEX 3	✓	✓	✓	✓	✓	✓	✓	✓	✓ *	Cre14.g618450	non-specific serine/threonine protein kinase
PEX 4	✓	✓	✓	✓				✓	✓	Cre05.g240150	ubiquitin-conjugating enzyme e2-21 kda
PEX 5	✓	✓	✓	✓	✓	✓	✓	✓	✓	Cre14.g616750	peroxisomal targeting signal 1 receptor PEX5
PEX 6	✓	✓	✓	✓	✓	✓	✓	✓	✓	Cre10.g446250	peroxisome assembly factor 2
PEX 7	✓	✓	✓	✓	✓	✓	✓	✓	✓	Cre17.g730200	peroxisomal targeting signal 2 receptor PEX7
PEX 8	✓	✓	✓	✓							
PEX 10	✓	✓	✓	✓	✓	✓	✓	✓	✓	Cre07.g337050	peroxisome assembly protein 10
PEX 11	✓	✓	✓	✓	✓	✓	✓	✓	✓	Cre06.g263300	peroxisomal biogenesis factor 11
PEX 12	✓	✓	✓	✓	✓	✓	✓	✓	✓	Cre06.g263602	peroxisome assembly protein 12 peroxin-12
PEX 13	✓	✓	✓	✓	✓	✓	✓	✓	✓	Cre13.g574700	peroxisomal membrane protein PEX13
PEX 14	✓	✓	✓	✓	✓	✓	✓	✓	✓	Cre10.g454025	peroxisomal membrane protein PEX14
PEX 15	✓									no hits: Sc PEX15	
PEX 16	**	✓		✓	✓	✓	✓	✓	✓	Cre11.g467758	peroxisomal membrane protein PEX16
PEX 17	✓	✓	✓					✓ ***		no hits: Sc PEX or At PUP7	
PEX 18	✓										
PEX 19	✓	✓	✓	✓	✓	✓	✓	✓	✓	Cre02.g119000	peroxisomal farnesylated protein
PEX 20		✓	✓	✓	✓					no hits: Sc and Nc PEX20	
PEX 21	✓									no hits: Sc PEX21	
PEX 22	✓	✓	✓	✓				✓	✓	Cre06.g283150	
PEX 23/30	✓	✓	✓	✓	✓		✓				
PEX 24/28	✓	✓	✓	✓							
PEX 25	✓	✓	✓								
PEX 26		✓ *	✓	✓		✓	✓			no hits: Hs or Mm PEX26	
PEX 27	✓								✓	Cre12.g540500	peroxisomal membrane protein PMP27
PEX 29	✓	✓	✓							no hits: Sc PEX29	
PEX 31	✓									no hits: Sc PEX31	
PEX 32	✓		✓							no hits: Sc PEX32	
PEX 33				✓						no hits: Sc PEX33	
PEX 34	✓									no hits: Sc PEX34	
	Smith and Aitchison (2013)								This work		
								Nito et al., (2007)			

## Data Availability

**T**he datasets for this study can be available upon request to N.K. The data presented in this study are openly available at https://phytozome-next.jgi.doe.gov or available upon request to the cor-responding author.

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
