# Peer review of "Chlamydomonas reinhardtii Alternates Peroxisomal Contents in Response to Trophic Conditions"

_cells, 2022, doi:10.3390/cells11172724_

Round 1

Reviewer 1 Report

In this work, Kato and coworkers suggest that C. reinhardtii has a complete set  of peroxisome biogenesis factors and present a list of potential Chlamydomonas peroxins (PEXs) required for peroxisomal biogenesis. Also, they examine involvements of the peroxisomes in the metabolism  of exogenously added fatty acids. They use transgenic Chlamydomonas expressing fluorescent proteins fused to peroxisomal proteins, together with fluorescently labeled palmitic acid under different carbon sources. Their results suggest that lipid biosynthesis but not β-oxidation is the primary metabolic preference of fatty acids in the microalga. Although lipid metabolism is an important topic that needs to be further explored in this organism, more experimental data should be provided in order to the novelty of this paper. Below you can find some suggestions:

1. Authors should show the expression levels of the peroxisome biogenesis genes that they put together in Chlamydomonas under the different conditions they tested along the manuscript (light, dark, with or without acetate). This could add a lot of valuable information for them to discuss.

2. Authors also claim that Chlamydomonas is not able to grow under fatty acids used as C supply in the dark. This is should be explain in more detail, I mean why in first place they think fatty acids can be used as (solely) external carbon source in this alga and what is presumably the uptake mechanism.

3. Authors used co-localization of different fluorescent constructs expressed in Chlamydomonas to determine the different content of peroxisomes. I wonder whether authors have also used TEM to identify these structures. I think putting together TEM and colocalization should help in defining these structures in the algal cells.

As minor aspects, several editing errors are present in this version of the manuscript (several writing type for instance), author should check on this.

Author Response

Please find an uploaded PDF file. 

Reviewer 2 Report

The manuscript by Kato et al. describes some features of the poorly characterised peroxisomes of Chlamydomonas reinhardtii. Using data of growth from fatty acids, bioinformatics search of peroxisomal proteins and confocal microscopy, the authors conclude that Chlamydomonas peroxisomes have only a minor role (if any) in beta oxidation of fatty acids. This conclusion is supported by (1) results showing that fatty acids cannot support growth as sole carbon sources, (2) there is low co-localization of a fatty acid species and peroxisomal proteins. The reduced/null beta oxidation occurs even when the authors find that Chlamydomonas is equipped with a near complete set of peroxisome biogenesis proteins. 

The manuscript is well written and the figures have the appropriate quality. However, in my opinion there are some points that need to be addressed in order to be acceptable for publication: 

Title

-The title of the manuscript (“Chlamydomonas reinhardtii alternates peroxisomal contents in response to light conditions”) refers to the observation made by the authors of the effect of light on the co-localization of a glyoxylate cycle enzyme marker, a protein containing the peroxisomal targeting signal 1 peptide and a fatty acid marker. The authors find that the effect of light in co-localization frequency of the three markers when there is acetate in the medium is rather negligible, while when cells have been transferred to a medium without acetate, light imparts a more dramatic change in the co-localization distribution of the three markers. Therefore, the title of this work seems misleading because it is not light itself but the metabolic state of the cell which seems to have an effect on peroxisome content. On the other hand, the cells analyzed in this experiment were transferred from TAP media in continuous light to minimal media (no acetate), with and without light, for 24h before imaging. 24h is probably a short period of time for the acclimation of cells to the new growing conditions, and the observed change in peroxisome composition could be part of the response to a shock and/or cells transitioning to a new metabolic state. Therefore, it is hard to determine what is driving the change in marker localization. Please comment. 

Authors analyse a fair amount of cells to quantify marker co-localization, and their results would benefit from some kind statistical analysis that could rule out the possibility that the change in peroxisome composition is not simply a consequence of reducing the number of peroxisomes per cell (Figure 3C).

Introduction 

-line 45 and 49, authors briefly describe the metabolism of acetate, implying that this happens in peroxisomes based on previous work (by the same authors) that has shown peroxisome localization of some of the enzymes involved in its metabolism. This view contrasts with metabolic work that shows that acetate metabolism occurs in mitochondria, chloroplast and cytosol. Could you please clarify and add the appropriate text to the introduction? 

Material and Methods, Results

-the data presented in Figure 1 is used by the authors to conclude that C. reinhardtii is not able to grow on exogenous fatty acids. The right panels of Figure 1A show growth curves of cultures incubated with acetate, butyric acid and valeric acid, plus control (tween 80 alone) for two different strains. The cell density of the culture was determined for only three days after transfer to the test media. Culture was incubated at 23C, which results in a doubling time longer than 24h for the acetate containing media (in dark). If the other compounds were supporting growth but at a slower rate than acetate, the increase in cell number would have been missed in such a short time course. In addition, it is possible that there is a lag phase at the beginning of the time course because cells would be adapting to a potential new carbon source. If authors want to conclude that the compounds tested cannot support growth, they should present an extended time course. 

-Authors use Tween 80 as a dispersal agent for fatty acids in the growth experiments shown in Figure 1, arguing that it “has been previously shown to be effective for this purpose in other microbial studies”. Is there any previous work that shows that Tween 80 can be used for this purpose for Chlamydomonas cultures? Can the authors show (and include in the manuscript) that fatty acids actually are being uptaken by the cells?

-Please include number of technical/biological replicates which are presented in Figure 1. 

-Figure 1C, cell density in glucose with no palmitic acid in light and acetic acid with palmitic acid in dark are anomalously low. How reproducible were these results? Can authors hypothesise reasons behind the growth inhibition?

-Authors present a table of putative peroxisome biogenesis genes that they identify in Chlamydomonas (Table 1). The description of the bioinformatics strategy to identify these genes is poor, authors should add more info about how this was done. Identity values with characterised genes in other species will strengthen their conclusions. Is there any experimental data about any of the Chlamydomonas genes listed in this table that support a role in peroxisome biogenesis?

-Figure 2, bottom panel, Does the CIS2-CFP fusion protein expresses the full length protein or only the 21 first amino acids, as in CIS2-RFP? Could you please comment on this panel in the main text? 

-Authors should mention in the main text that the co-localization experiments were done using UVM4 cells, if that was the case, and comment that the marker localization that they observe may not apply to wild type cells, where the level of expression of these markers would be much lower. In addition, UVM4 was generated by UV mutagenesis, and the localization pattern may be specific to the genetic background of the mutant. 

Discussion

-paragraph about peroxisomal function and evolution in green alga. I am a bit confused by the author’s argument that “phototaxis may have meant that these organisms do not need to rely on beta-oxidation for energy generation in the dark”, which seems plausible. However, this hypothesis is not supported by the fact that Chlorella, which is non-flagellated, does not have beta-oxidation either. could the authors please clarify?

In addition, there are some typos and editing issues: 

-Line 234, the last sentence is not complete. 

-line 305, there is citation of a Table 16, which does not exist in the manuscript, and a paragraph that seems to be the last paragraph of the results section. 

-supplementary data requires improved presentation and the addition of more information: there is no title, no headlines, there is no connection between the first two paragraphs, or a reason why they are there.  

-line 396, typo in acetate

-reference list: I noted an error in the reference Young and Shacchar. The year is 2021 and not 2020 as it is printed. 

Author Response

Please find an uploaded PDF file. 

Round 2

Reviewer 1 Report

The authors have highly improved the information given compared with the first version of this manuscript and have adequately addressed all my suggestions and comments. 

Reviewer 2 Report

Authors have addressed most of points raised in the first review.